# Synthesis of 5-Benzylamino and 5-Alkylamino-Substituted Pyrimido[4,5-c]quinoline Derivatives as CSNK2A Inhibitors with Antiviral Activity

**DOI:** 10.3390/ph17030306

**Published:** 2024-02-27

**Authors:** Kesatebrhan Haile Asressu, Jeffery L. Smith, Rebekah J. Dickmander, Nathaniel J. Moorman, James Wellnitz, Konstantin I. Popov, Alison D. Axtman, Timothy M. Willson

**Affiliations:** 1Structural Genomics Consortium, UNC Eshelman School of Pharmacy, University of North Carolina at Chapel Hill, Chapel Hill, NC 27599, USA; kesateha@unc.edu (K.H.A.); jeffs@email.unc.edu (J.L.S.);; 2Rapidly Emerging Antiviral Drug Development Initiative (READDI), Chapel Hill, NC 27599, USA; 3Department of Microbiology and Immunology, University of North Carolina at Chapel Hill, Chapel Hill, NC 27599, USA; 4Lineberger Comprehensive Cancer Center, University of North Carolina at Chapel Hill, Chapel Hill, NC 27599, USA; 5Department of Chemistry, University of North Carolina at Chapel Hill, Chapel Hill, NC 27599, USA; 6Laboratory for Molecular Modeling, Division of Chemical Biology and Medicinal Chemistry, UNC Eshelman School of Pharmacy, University of North Carolina at Chapel Hill, Chapel Hill, NC 27599, USA; 7Center of Integrative Chemical Biology and Drug Discovery, UNC Eshelman School of Pharmacy, University of North Carolina at Chapel Hill, Chapel Hill, NC 27599, USA

**Keywords:** CSNK2A, kinase inhibitor, structure–activity study, antiviral, conformational analysis

## Abstract

A series of 5-benzylamine-substituted pyrimido[4,5-c]quinoline derivatives of the CSNK2A chemical probe SGC-CK2-2 were synthesized with the goal of improving kinase inhibitor cellular potency and antiviral phenotypic activity while maintaining aqueous solubility. Among the range of analogs, those bearing electron-withdrawing (**4c** and **4g**) or donating (**4f**) substituents on the benzyl ring as well as introduction of non-aromatic groups such as the cyclohexylmethyl (**4t**) were shown to maintain CSNK2A activity. The CSNK2A activity was also retained with *N*-methylation of SGC-CK2-2, but α-methyl substitution of the benzyl substituent led to a 10-fold reduction in potency. CSNK2A inhibition potency was restored with indene-based compound **4af**, with activity residing in the S-enantiomer (**4ag**). Analogs with the highest CSNK2A potency showed good activity for inhibition of Mouse Hepatitis Virus (MHV) replication. Conformational analysis indicated that analogs with the best CSNK2A inhibition (**4t**, **4ac**, and **4af**) exhibited smaller differences between their ground state conformation and their predicted binding pose. Analogs with reduced activity (**4ad**, **4ae**, and **4ai**) required more substantial conformational changes from their ground state within the CSNK2A protein pocket.

## 1. Introductions

Casein Kinase 2 (CSNK2) is a constitutively active serine/threonine kinase that plays a key role in multiple cellular processes including viral entry [1,2]. CSNK2 is found in cells as a heterotetramer composed of two catalytic units (CSNK2A1 and CSNK2A2) and two regulatory units (dimer of CSNK2B). The pyrazolo[1,5-a]pyrimidine SGC-CK2-1 is a high-quality chemical probe for CSNK2A that demonstrated cellular target engagement at 20 nM and had remarkable selectivity across the DiscoverX KinomeSCAN assay panel of >400 human kinases, with 90% inhibition of only CSNK2A1, CSNK2A2, and STK17B at 1 µM [3]. Silmitasertib (CX-4945) is a widely studied CSNK2A inhibitor from the benzo[c][2,6]naphthyridine chemotype that is in Phase 2 clinical development for treatment of rare cancers [4,5]. Compared to SGC-CK2-1, silmitasertib was about 10-fold less potent as an inhibitor of CSNK2A in cells and much less selective across the kinome (Table 1). However, despite the lower potency and selectivity, silmitasertib has good aqueous solubility and pharmacokinetics that has allowed it to be widely used for pharmacological studies of CSNK2A in vivo. Recently, we identified the pyrimido[4,5-c]quinoline SGC-CK2-2 (Analog-2) [6] as a 4-aza analog of silmitasertib with much improved kinase selectivity. Although SGC-CK2-2 demonstrated nanomolar in-cell target engagement of CSNK2A (Table 1), it had only micromolar potency in antiviral phenotypic assays, such as the inhibition of β-coronavirus (β-CoV) replication (MHV IC_50_ = 15 µM). Our initial structure–activity studies of the pyrimido[4,5-c]quinoline 5-amino substituent identified the naphthylamine (Analog-11) with an increased cellular potency on CSNK2A, but it had lower aqueous solubility than SGC-CK2-2 and was less selective across the DiscoverX kinase panel (Table 1) [6]. Notably, in the prior study, SGC-CK2-2 was the singular example of a benzylamino substituent at the 5 position of the pyrimido[4,5-c]quinoline core. The X-ray structure of SGC-CK2-2 in CSNK2A (Figure 1) showed the benzyl group in a rotated conformation compared to the aryl group of silmitasertib, which changed the orientation of the π-stacking interaction with His160 [6]. Based on these differences we decided to perform an expanded structure–activity study to determine if a modified benzylamine analog could be identified with improved CSNK2A potency and antiviral activity while retaining good aqueous solubility.

## 2. Results and Discussion

### 2.1. General Synthetic Method for 5-Amino-Substituted Pyrimido[4,5-c]quinoline Analogs

Counterparts of SGC-CK2-2 bearing substituted 5-benzylamino substituents were designed to explore the nature of the interaction with His160 observed in the X-ray cocrystal structure with CSNK2A (Figure 1). The synthesis of 5-benzylamino analogs used modifications of the previously reported synthetic methods (Figure 1) [6]. Nucleophilic aromatic substitution (S_N_Ar) reactions of commercially available pyrimido[4,5-c]quinoline methyl ester **1** with 2-, 3-, and 4-substituted benzylamines **2a**–**m** afforded the corresponding naphthyridine analogs (**3a**–**m**). LiOH-mediated ester hydrolysis in MeOH afforded the free carboxylic acids (**4a**–**m**, Table 2) in a 42–92% overall yield.

We also explored the S_N_Ar reaction of **1** with amines **2n**–**p** containing pyridylmethyl in place of the benzyl ring (Figure 2). The resulting pyridylmethyl analogs **3n**–**p** were saponified to the corresponding carboxylic acids (**4n**–**p**, Table 3) in a 46–85% overall yield.

Synthesis of 5-*N*-alkylamino analogs utilized the S_N_Ar reaction of **1** with aliphatic primary amines **2q**–**ab** and **2ae**–**ah** or secondary amines **2ac**–**ad** and **2ai** (Figure 3), which included both *N*-methylated and conformationally constrained benzylamines. Saponification of the methyl esters afforded the carboxylic acids (**4q**–**ai**, Table 4 and Table 5) in a 35–98% overall yield.

### 2.2. CSNK2A Potency and Aqueous Solubility of 5-Benzylamino Analogs

The analogs were evaluated for in-cell CSNK2A target engagement by NanoBRET assay and for aqueous solubility. The NanoBRET data showed that the new 5-benzylamino analogs had CSNK2A cellular activity with IC_50_ ≤ 1 μM (Table 2, Appendix A). Notably, the SAR revealed that addition of small electron donating or withdrawing substituents to the aromatic ring (**4a**–**m**) did not have any dramatic effect on CSNK2A activity. These results suggested that there was unlikely to be a strong π-stacking interaction between the benzyl amine and His160 in the CSNK2A pocket. The kinetic solubility of analogs **4a**–**k** was found to be slightly lower than SGC-CK2-2, but still >100 μM. The only analogs with low solubility were the lipophilic biphenyl (**4l**) and naphthalene (**4m**) substituted amines.

Replacement of the phenyl ring with pyridine in analogs **4n**–**p** revealed that the CSNK2A cellular potency was greatly reduced (Table 3) indicating that the heterocycle was not tolerated in the binding pocket. However, the 3-pyridyl (**4o**) and 4-pyridyl (**4p**) analogs displayed improved aqueous solubility.

### 2.3. CSNK2A Activity and Aqueous Solubility of Non-Aromatic Amine-Based Analogs

Having demonstrated the absence of a strong π-stacking interaction with His160, we decided to synthesize analogs with lipophilic non-aromatic amines including aliphatic and alicyclic amines with variable ring sizes, as well as bridged ring systems (Table 4). The aliphatic isopropyl analog **4q** was found to be less potent (IC_50_ = 1.2 μM) than SGC-CK2-2. However, potency was increased 3-fold with cyclic analogs **4r** and **4s** (IC_50_ = 0.3–0.4 μM). The cyclohexylmethyl analog (**4t**) was shown to have the optimal ring size with a CSNK2A IC_50_ = 0.14 μM. An equivalent CSNK2A cellular potency was obtained with other 5-alkylamino analogs containing larger rings (**4u**) and bridged rings (**4w** and **4x**) (Table 4). The adamantyl-substituted analog **4x** maintained CSNK2A potency, but at a cost of lower aqueous solubility (10 μM).

Introduction of a oxygen heteroatom into the aliphatic ring (**4y** and **4z**) resulted in a 10-fold decrease in CSNK2A potency compared to the parent compounds (**4s** and **4w**). Incorporation of difluoro groups in analog **4aa** maintained cellular potency, whereas introducing a nitrogen heteroatom (**4ab**) greatly reduced the CSNK2A activity.

### 2.4. CSNK2A Activity of Structurally and Conformationally Constrained Analogs

Several analogs were synthesized to explore the effect of constraining the conformation of the 5-benzyl substituent (Table 5). CSNK2A cellular activity did not change with *N*-methylation of the benzyl group (**4ac**), but surprisingly, the corresponding *N*-methylated cyclohexylmethylamine (**4ad**) was 17-fold less active than its unmethylated parent amine (**4t**). α-Methyl substitution (**4ae**) on the benzylamine of SGC-CK2-2 also led to a reduction in CSNK2A potency (IC_50_ = 1.9 µM). Interestingly, the 2,3-dihydro-1*H*-indene (**4af**), which constrained the methylene substituent in a ring fusion, had equivalent activity to SGC-CK2-2 (IC_50_ = 0.23 µM). In contrast, the 1,2,3,4-tetrahydroisoquinoline (**4ai**) which constrained the *N*-methyl substituent in a piperidine ring was much less active (Table 5). Since potent inhibitor **4af** has a chiral center, we synthesized the *R*- and *S*-isomers from the corresponding enantiomerically pure amines. CSNK2A activity was retained exclusively in the *S*-enantiomer (**4ag**). All compounds in this series had good aqueous solubility (>150 μM).

To probe the potential contribution of conformational constraint to explain the SAR of the *N*-methyl and α-methyl-substitutions, we performed an analysis of the ground state and the bound pose for four “active” analogs (SGC-CK2-2, **4t**, **4ac**, and **4af**) and three “inactive” analogs (**4ad**, **4ae**, and **4ai**) with 10-fold differences in IC_50_. The ground state was computed as the lowest energy conformation for each analog in water (see Section 3.4). The computational modelling suggested that the low-energy poses of the active analogs (**4t**, **4ac**, and **4af**) placed their ring density in the same place as SGC-CK2-2, while their inactive counterparts (**4ad**, **4ae**, and **4ai**) preferred a low-energy pose with their ring density closer to the naphthyridine core (Figure 2A). To model the potential bound pose within the ATP-binding pocket of CSNK2A, we aligned the ground state conformation of each analog with the crystallographically determined pose of SGC-CK2-2 (PDB: 8BGC) [6] and subsequently relaxed it within the protein binding site. The conformational analysis indicated that analogs with higher CSNK2A activity (**4t**, **4ac**, and **4af**) exhibited smaller differences between the ground state conformation and the predicted binding pose (Figure 2B). The computational modeling suggested that the less active analogs (**4ad**, **4ae**, and **4ai**) required more substantial conformational change from their ground state to assume the optimal bound pose within the CSNK2A enzyme pocket (Figure 2B). The root-mean-square deviation (RMSD) of atomic positions was used to quantify conformational change between the ground state and predicted bound pose, with the three inactive analogs requiring a larger repositioning of their atoms to fit into the CSNK2A pocket (Table 6).

### 2.5. Antiviral Activity of Naphthyridine Analogs

The antiviral activity of the CSNK2A inhibitors with IC_50_ < 0.2 µM was determined on Mouse Hepatitis Virus (MHV), a member of the *β*-coronavirus (*β*-CoV) genus [7], using an MHV-NLuc assay in mouse derived-from-brain-tumor (DBT) cells [6,8]. All CSNK2A active compounds except the 1-adamantyl analog **4x** demonstrated inhibition of viral replication (Table 7 and Appendix A). The antiviral activity of these analogs was in the same potency range as SGC-CK2-2 (IC_50_ = 10–20 µM) indicating that the modification of the 5-benzyl substituent was well tolerated but failed to significantly improve phenotypic activity. The loss in activity of compound **4x** is likely due to its poor aqueous solubility (~10 µM).

## 3. Materials and Methods

### 3.1. NanoBRET Assay

HEK293 cells were cultured at 37 °C, 5% CO_2_ in Dulbecco’s modified Eagle medium (DMEM; Gibco, New York, NY, USA) supplemented with 10% fetal bovine serum (VWR/Avantor, Allentown, PA, USA). A transfection complex of DNA at 10 µg/mL was created, consisting of 9 µg/mL of carrier DNA (Promega, Madison, WI, USA) and 1 µg/mL of CSNK2A2-NLuc fusion DNA in Opti-MEM without serum (Gibco). FuGENE HD (Promega) was added at 30 µL/mL to form a lipid/DNA complex. The solution was then mixed and incubated at room temperature for 20 min. The transition complex was mixed with a 20× volume of HEK293 cells at 20,000 cells per mL in DMEM/FBS and 100 µL per well was added to a 96-well plate that was incubated overnight at 37 °C, 5% CO_2_. The following day, the media was removed via aspiration and replaced with 85 μL of Opti-MEM without phenol red. A total of 5 μL per well of 20×-NanoBRET Tracer K10 (Promega) at 5 μM in Tracer Dilution Buffer (Promega N291B) was added to all wells, except the “no tracer” control wells. Test compounds (10 mM in DMSO) were diluted 100× in Opti-MEM media to prepare stock solutions and evaluated at eleven concentrations. A total of 10 μL per well of the 10-fold test compound stock solutions (final assay concentration of 0.1% DMSO) was added. For “no compound” and “no tracer” control wells, DMSO in OptiMEM was added for a final concentration of 1.1% across all wells. 96-well plates containing cells with NanoBRET Tracer K10 and test compounds (100 µL total volume per well) were equilibrated (37 °C/5% CO_2_) for 2 h. The plates were cooled to room temperature for 15 min. NanoBRET NanoGlo substrate (Promega) at a ratio of 1:166 to Opti-MEM media in combination with extracellular NLuc Inhibitor (Promega) diluted 1:500 (10 μL of 30 mM stock per 5 mL Opti-MEM plus substrate) were combined to create a 3× stock solution. A total of 50 μL of the 3× substrate/extracellular NL inhibitor were added to each well. The plates were read within 30 min on a GloMax Discover luminometer (Promega) equipped with 450 nm BP filter (donor) and 600 nm LP filter (acceptor) using 0.3 s integration time. Raw milliBRET (mBRET) values were obtained by dividing the acceptor emission values (600 nm) by the donor emission values (450 nm) and multiplying by 1000. Averaged control values were used to represent complete inhibition (no tracer control: Opti-MEM + DMSO only) and no inhibition (tracer only control: no compound, Opti-MEM + DMSO + Tracer K10 only) and were plotted alongside the raw mBRET values. The data were first normalized and then fit using Sigmoidal, 4PL binding curve in GraphPad Prism version 9.5.1 to determine IC_50_ values (Appendix A).

### 3.2. MHV Assay

DBT cells were cultured at 37 °C in Dulbecco’s modified Eagle medium (DMEM) (MilliporeSigma, St. Louis, MO, USA) supplemented with 10% fetal bovine serum (Gibco, New York, NY, USA) and penicillin and streptomycin (MilliporeSigma, St. Louis, MO, USA). DBT cells were plated in 96-well plates to be 80% confluent at the start of the assay. Test compounds were diluted to 30 µM in DMEM. Serial 4-fold dilutions were made in DMEM, providing a concentration range of 30 µM to 0.0018 µM. Molnupiravir (10 µM) was included as a positive control. Media was aspirated from the DBT cells and 100 μL of the diluted test compounds were added to the cells for 1 h at 37 °C. After 1 h, MHV-NLuc was added at an MOI of 0.1 in 50 μL DMEM so that the final concentration of the first dilution of compound was 30 μM (T = 0). After 10 h, the media was aspirated, and the cells were washed with PBS and lysed with passive lysis buffer (Promega) for 20 min at room temperature. Relative light units (RLUs) were measured using a luminometer (Promega; GloMax). Triplicate data were analyzed in GraphPad Prism version 9.5.1 to generate IC_50_ values (Appendix A).

### 3.3. Kinetic Solubility

Kinetic solubility assay was conducted using 10 mM DMSO stocks of compounds in phosphate-buffered saline solution (PBS) at pH 7.4 by Analiza, Inc., Cleveland, OH, USA. Following 24 h incubation in PBS, total chemiluminescent nitrogen determination was used to quantify the solubility of the compounds against compound standards. The calculated values were corrected for the background nitrogen in the media and DMSO.

### 3.4. Molecular Modeling

Analogs **4t**, **4ac**, **4ad**, **4ae**, **4af**, and **4ai** were assigned initial geometries using RDKit version 2022.3.5. Geometries were optimized using density functional theory in the Gaussian16 software package with a B3LYP level of theory, 6-311 + g(d) basis set, and a CPCM water solvent model [9]. Optimized geometries were locked and rigidly aligned to the crystal pose of the parent compound SGC-CK2-2 in CSNK2A (PDB: 8BGC) [6] using the Schrödinger software suite version 2023-2. The protein structure of the CSNK2A (PDB: 8BGC) was prepared using the standard settings of Schrödinger protein preparation wizard to provide minor structure optimization and ensure optimal performance of the Glide docking. The positions of the hydrogen atoms were optimized, residue ionization and tautomer states were checked using PROPKA version 3.5.0 at pH 7, water molecules further than 3 Å away from the HET atoms were removed, and restrained minimization of the structure was performed with the OPLS4 force field, converging the heavy atoms to an RMSD of 0.3 Å. The resulting structure was used to generate the Glide receptor grid using the center of mass of the X-ray bound ligand with the inner and outer box sizes equal to 10 Å and 20 Å, respectively. To obtain the aligned poses of the analogs, each one was minimized in place, considering the protein environment, using the Extra precision refinement mode in Glide [10,11].

### 3.5. Synthetic Procedures

#### 3.5.1. General Information

The reagents and solvents were used without any further purification after they were received from commercial suppliers. Methyl 5-chloropyrimido[4,5-c]quinoline-8-carboxylate (**1**) (catalog number EN300-26862632) was purchased from Enamine, Kyiv, Ukraine. Unless otherwise specified in degrees Celsius (°C), reactions were conducted under an argon atmosphere and at ambient temperature. Analytical Aluminum thin-layer chromatography (Al TLC) (performed on pre-coated fluorescent silica gel plates, 200 μm with an F254 indicator) and LC-MS were used to monitor the progress of reaction mixtures and the purity of fractions collected from flash column chromatography. A rotary evaporator was used to remove the solvent under reduced pressure (in vacuo). Following Al TLC separation, the position of each spot was detected using UV light (254/365 nm). Nuclear magnetic resonance (NMR) spectra and microanalytical data were collected for final compounds to confirm their identity and assess their purity. ^1^H and ^13^C NMR spectra were recorded in DMSO-*d*_6_ on a Avance 400 and 500 MHz (Bruker, Billerica, MA, USA). Chemical shifts are reported in parts per million (ppm), with residual solvent peaks referenced as the internal standard. Coupling constants (*J* values) are expressed in hertz (Hz). Spin multiplicities are described as singlet (s), broad singlet (bs), doublet (d), doublet of doublets (dd), triplet (t), quartet (q), pentent (p), and multiplet (m). Data were processed using MestReNova version 14.3.1-31739. High-resolution mass spectrometry samples were analyzed with a ThermoFisher Q Exactive HF-X (ThermoFisher, Bremen, Germany) mass spectrometer coupled with a Waters Acquity H-class liquid chromatograph system. Samples were introduced via a heated electrospray source (HESI) at a flow rate of 0.3 mL/min. Electrospray source conditions were set as follows: spray voltage 3.0 kV, sheath gas (nitrogen) 60 arb, auxiliary gas (nitrogen) 20 arb, sweep gas (nitrogen gas) 0 arb, nebulizer temperature 375 °C, capillary temperature 380 °C, RF funnel 45 V. The mass range was set to 150–2000 *m*/*z*. All measurements were recorded at a resolution setting of 120,000. Separations were conducted on a Waters Acquity UPLC BEH C18 column (2.1 × 50 mm, 1.7 μm particle size). LC conditions were set at 95% water with 0.1% formic acid (A) ramped linearly over 5.0 min to 100% acetonitrile with 0.1% formic acid (B) and held until 6.0 min. At 7.0 min, the gradient was switched back to 95% A and allowed to re-equilibrate until 9.0 min. Injection volume for all samples was 3 μL. Xcalibur (ThermoFisher, Breman, Germany) was used to analyze the data. Solutions were analyzed at 0.1 mg/mL or less based on responsiveness to the ESI mechanism. All observed species were singly charged, as verified by unit *m*/*z* separation between mass spectral peaks corresponding to the ^12^C and ^13^C^12^Cc-1 isotope for each elemental composition. Purity of synthesized of compounds was determined using NMR or LC–MS. All final compounds were >95% pure except for **4n** (92%) and **4p** (92%).

#### 3.5.2. General Synthetic Methods

To the solution of methyl 5-chloropyrimido[4,5-c]quinoline-8-carboxylate (**1**, 40 mg, 1 eq, 0.15 mmol) in anhydrous 1-methyl-2-pyrrolidinone (NMP) (0.7 mL), amine (**2**, 21–41 mg, 2 eq, 0.29 mmol), and *N*,*N*-diisopropylethylamine (DIPEA) (20 mg, 27 μL, 1.05 Eq, 0.15 mmol) were added. The reaction mixture was sealed and stirred for 1–2 h at 100 °C. The crude reaction mixture was cooled to rt and analyzed by Al TLC using 5% MeOH in DCM and by LC-MS. After the consumption of the ester starting material and formation of the desired product confirmed by those analyses, the crude reaction mixture was diluted with H_2_O to remove the polar NMP solvent. The crude product was extracted with ethyl acetate (EA) from the aqueous (aq) phase. The EA layer was washed three times with H_2_O to remove residual NMP. The organic phase was dried in vacuo to yield the crude product, which was purified using a Biotage flash column chromatography on a 12 g column using a DCM/MeOH solvent system. The desired product was obtained in the 3–6% MeOH/DCM range. The fractions which contained the desired product by TLC or LC-MS, were combined and concentrated in vacuo to yield the pure methyl ester **3** as a pale-yellow solid. The methyl ester **3** (1 eq, 94 μmol) was suspended in MeOH (0.6 mL) and aq. 2M LiOH (0.71 mL, 15 eq, 1.4 mmol) was added. The resulting mixture was stirred at 45–70 °C for 4–15 h. The progress of the reaction was analyzed by TLC using 5–10% MeOH in DCM and LC-MS. After confirming the consumption of the ester intermediate and complete formation of the desired product, the reaction mixture was evaporated to remove the MeOH solvent, and the crude product was dissolved in water and 1N HCl added until the pH reached 3.5. The milky solution was transferred to an eppendorf tube (1.5 mL in size) and spun in a centrifuge for 3 min to precipitate the carboxylic acid. The precipitated product was washed three times with water to remove residual LiCl salt. The carboxylic acid **4** was dried in air and then under high vacuum, and the final yield was calculated. The purity of the final product **4** was confirmed by LC-MS and NMR analysis (Appendix A).

5-((2-chlorobenzyl)amino)pyrimido[4,5-c]quinoline-8-carboxylic acid (**4a**)

Yellow solid (62 mg, 92%), mp: 193–198 °C; ^1^H NMR (400 MHz, DMSO-*d*_6_) δ (ppm) 10.28 (s, 1H), 9.50 (s, 1H), 8.53 (d, *J* = 12.0 Hz, 1H), 8.43 (t, *J* = 8.0 Hz, 1H), 8.07 (s, 1H), 7.89 (d, *J* = 8.0 Hz, 1H), 7.48–7.43 (m, 2H), 7.26–7.24 (m, 2H), 4.90 (d, *J* = 8.0 Hz, 2H). ^13^C NMR (212.5 MHz, DMSO- *d*_6_) δ 168.20, 155.90, 155.71, 152.14, 144.23, 143.07, 138.13, 137.00, 132.11, 129.04, 128.72, 128.27, 127.09, 127.00, 124.39, 124.29, 121.10, 117.11, 41.58. Purity: 100%, HRMS (ESI) calculated for C_19_H_14_N_4_O_2_Cl [M + H]: 365.0700, found: 365.0803.

5-((3-chlorobenzyl)amino)pyrimido[4,5-c]quinoline-8-carboxylic acid (**4b**)

Yellow solid (48 mg, 70%), mp: 240–243 °C; ^1^H NMR (400 MHz, DMSO- *d*_6_) δ (ppm) 13.07 (br s, 1H), 10.32 (s, 1H), 9.56 (s, 1H), 8.83 (t, *J* = 8.8 Hz, 1H), 8.73 (d, *J* = 8.72 Hz, 1H), 8.12 (s, 1H), 7.85 (d, *J* = 8.74 Hz, 1H), 7.53 (t, *J* = 4.0 Hz, 1H), 7.45 (dt, *J* = 4.0, 8.0 Hz, 1H), 7.34 (t, *J* = 8.0 Hz, 1H), 7.28–7.25 (m, 1H), 4.82 (d, *J* = 8.0 Hz, 2H). ^13^C NMR (212.5 MHz, DMSO-*d*_6_) δ 167.11, 156.76, 156.40, 152.94, 144.44, 142.60, 139.07, 132.82, 132.23, 130.06, 127.52, 127.45, 126.59, 126.40, 123.86, 123.18, 122.57, 120.42, 43.11. Purity: 95%, HRMS (ESI) calculated for C_19_H_14_N_4_O_2_Cl [M + H]^+^: 365.0700, 365.0810.

5-((4-chlorobenzyl)amino)pyrimido[4,5-c]quinoline-8-carboxylic acid (**4c**)

Yellow solid (31 mg, 76%), mp: 203–205 °C; ^1^H NMR (400 MHz, DMSO-*d*_6_) δ (ppm) 13.19 (br s, 1H), 10.31 (s, 1H), 9.55 (s, 1H), 8.80 (t, *J* = 6.3 Hz, 1H), 8.71 (d, *J* = 8.5 Hz, 1H), 8.11 (d, *J* = 1.7 Hz, 1H), 7.85 (dd, *J* = 8.3, 1.8 Hz, 1H), 7.50 (d, *J* = 8.5 Hz, 2H), 7.36 (d, *J* = 8.5 Hz, 2H), 4.80 (d, *J* = 8.0 Hz, 2H). ^13^C NMR (126 MHz, DMSO-*d*_6_) δ 167.17, 156.73, 156.38, 152.91, 144.48, 139.05, 139.00, 138.99, 132.32, 131.17, 129.55, 128.12, 127.55, 123.84, 123.11, 122.55, 120.31, 42.97. Purity: 100%, HRMS (ESI) calculated for C_19_H_14_N_4_O_2_Cl [M + H]^+^: 365.0700, found: 365.0804.

5-((2-methoxybenzyl)amino)pyrimido[4,5-c]quinoline-8-carboxylic acid (**4d**)

Yellow solid (67 mg, 85%), mp: 230–233 °C; ^1^H NMR (500 MHz, DMSO-*d*_6_) δ (ppm) 10.32 (s, 1H), 9.56 (s, 1H), 8.72 (d, *J* = 10.0 Hz, 1H), 8.33 (t, *J* = 5.0 Hz, 1H), 8.09 (d, *J* = 1.6 Hz, 1H), 7.85 (dd, *J* = 5.0, 10.0 Hz, 1H), 7.27 (dd, *J* = 5.0, 10.0 Hz, 1H), 7.23 (td, *J* = 5.0, 10.0 Hz, 1H), 7.03 (td, *J* = 1.2, 8.4 Hz, 1H), 6.85 (td, *J* = 1.2, 5.0 Hz, 1H), 4.83 (d, *J* = 5.0 Hz, 2H), 3.89 (s, 3H). ^13^C NMR (212.5 MHz, DMSO-*d*_6_) δ 167.16, 156.92, 156.76, 156.40, 153.04, 144.55, 139.01, 132.34, 127.93, 127.62, 127.50, 126.81, 123.80, 123.12, 122.49, 120.27, 120.12, 110.55, 55.42, 39.69. Purity: 99%, HRMS (ESI) calculated for C_20_H_17_N_4_O_3_ [M + H]^+^: 361.1200, found: 361.1298.

5-((3-methoxybenzyl)amino)pyrimido[4,5-c]quinoline-8-carboxylic acid (**4e**)

Yellow solid (55 mg, 89%), mp: 210–214 °C; ^1^H NMR (400 MHz, DMSO-*d*_6_) δ (ppm) 10.31 (s, 1H), 9.54 (s, 1H), 8.70 (d, *J* = 8.0 Hz, 1H), 8.64 (t, *J* = 8.0 Hz, 1H), 8.13 (d, *J* = 1.8 Hz, 1H), 7.86 (dd, *J* = 4.0, 8.0 Hz, 1H), 7.21 (t, *J* = 8.0 Hz, 1H), 7.06–7.03 (m, 1H), 6.78 (dd, *J* = 2.6, 8.0 Hz, 1H), 4.79 (d, *J* = 4.0 Hz, 2H), 3.71(s, 3H). ^13^C NMR (212.5 MHz, DMSO-*d*_6_) δ 167.45, 159.22, 156.57, 156.31, 152.84, 144.53, 141.51, 138.94, 129.24, 127.43, 123.93, 122.85, 122.74, 113.54, 111.90, 54.91, 43.56. Purity: 97%, HRMS (ESI) calculated for C_20_H_17_N_4_O_3_ [M + H]^+^: 361.1200, found: 361.1298.

5-((4-methoxybenzyl)amino)pyrimido[4,5-c]quinoline-8-carboxylic acid (**4f**)

Yellow solid (40.7 mg, 68%), mp: 246–247 °C; ^1^H NMR (500 MHz, DMSO) δ 13.15 (br s, 1H), 10.30 (s, 1H), 9.53 (s, 1H), 8.70 (d, *J* = 8.4 Hz, 1H), 8.62 (t, *J* = 6.3 Hz, 1H), 8.14 (d, *J* = 1.7 Hz, 1H), 7.84 (dd, *J* = 8.2, 1.8 Hz, 1H), 7.47–7.39 (m, 2H), 6.87 (d, *J* = 8.6 Hz, 2H), 4.74 (d, *J* = 5.8 Hz, 2H), 3.70 (s, 4H). ^13^C NMR (126 MHz, DMSO-*d*_6_) δ 167.19, 158.16, 156.72, 156.35, 152.86, 144.61, 139.09, 139.06, 132.16, 131.76, 131.75, 129.11, 127.54, 123.80, 123.12, 122.39, 120.27, 113.60, 54.99, 43.05. Purity: 100%, HRMS (ESI) calculated for C_20_H_17_N_4_O_3_ [M + H]^+^: 361.1200, found: 361.1293.

5-((4-fluorobenzyl)amino)pyrimido[4,5-c]quinoline-8-carboxylic acid (**4g**)

Yellow solid (36 mg, 83%), mp: 235–238 °C; ^1^H NMR (400 MHz, DMSO-*d*_6_) δ (ppm) 13.14 (br s, 1H), 10.31 (s, 1H), 9.55 (s, 1H), 8.76 (t, *J* = 8.0 Hz, 1H), 8.72 (d, *J* = 8.0 Hz, 1H), 8.13 (d, *J* = 4.0 Hz, 1H), 7.85 (dd, *J* = 1.8, 8.3 Hz, 1H), 7.54–7.51 (m, 2H), 7.15–7.09 (m, 2H), 4.80 (d, *J* = 4.0 Hz, 2H). ^13^C NMR (100 MHz, DMSO-*d*_6_) δ 167.19, 162.32, 159.92, 156.74, 156.38, 144.53, 139.09, 136.10, 136.07, 132.21, 129.74, 129.66, 127.55, 123.84, 123.14, 122.50, 120.34, 114.98, 114.77, 42.90. Purity: 99.9%, HRMS (ESI) calculated for C_19_H_14_N_4_O_2_F [M + H]^+^: 349.1000, found: 349.1098.

5-((4-methylbenzyl)amino)pyrimido[4,5-c]quinoline-8-carboxylic acid (**4h**)

Yellow solid (32 mg, 79%), mp: 201–204 °C; ^1^H NMR (400 MHz, DMSO-*d*_6_) δ (ppm) 10.30 (s, 1H), 9.54 (s, 1H), 8.69 (d, *J* = 8.3 Hz, 1H), 8.63 (t, *J* = 6.3 Hz, 1H), 8.12 (d, *J* = 1.7 Hz, 1H), 7.84 (dd, *J* = 8.3, 1.8 Hz, 1H), 7.36 (d, *J* = 7.8 Hz, 2H), 7.11 (d, *J* = 7.8 Hz, 2H), 4.77 (d, *J* = 5.6 Hz, 2H), 2.24 (s, 3H). ^13^C NMR (100 MHz, DMSO-*d*_6_) δ 167.29, 156.65, 156.32, 152.86, 144.58, 139.02, 136.81, 136.80, 135.66, 132.89, 128.74, 127.67, 127.49, 123.84, 122.99, 122.51, 120.06, 43.35, 20.67. Purity: 100%. HRMS (ESI) calculated for C_20_H_17_N_4_O_2_ [M + H]^+^: 345.1300, found: 345.1349.

5-((4-(difluoromethyl)benzyl)amino)pyrimido[4,5-c]quinoline-8-carboxylic acid (**4i**)

Yellow solid (30.4 mg, 73%), mp: 250–255 °C; ^1^H NMR (400 MHz, DMSO-*d*_6_) δ (ppm) 13.10 (br s, 1H), 10.32 (s, 1H), 9.56 (s, 1H), 8.83 (t, *J* = 6.3 Hz, 1H), 8.71 (d, *J* = 8.0 Hz, 1H), 8.11 (d, *J* = 4.0 Hz, 1H), 7.85 (dd, *J* = 8.4, 1.8 Hz, 1H), 7.67–7.44 (m, 5H), 6.97 (t, *J* = 56.0 Hz, 1H), 4.87 (d, *J* = 5.9 Hz, 2H). ^13^C NMR (100 MHz, DMSO-*d*_6_) δ 167.17, 156.74, 156.38, 152.97, 144.48, 143.03, 139.07, 132.42, 132.23, 125.65, 125.59, 125.53, 123.84, 123.12, 122.54, 120.36, 117.33, 114.98, 112.64, 43.37. Purity: 100%. HRMS (ESI) calculated for C_20_H_15_N_4_O_2_F_2_ [M + H]^+^: 381.1100, found: 381.1161.

5-((4-(trifluoromethyl)benzyl)amino)pyrimido[4,5-c]quinoline-8-carboxylic acid (**4j**)

Yellow solid (38 mg, 86%), mp: 245–248 °C; ^1^H NMR (400 MHz, DMSO-*d*_6_) δ (ppm) 10.34 (s, 1H), 9.60 (s, 1H), 9.33 (br s, 1H), 8.75 (d, *J* = 8.0 Hz, 1H), 8.23 (s, 1H), 7.90 (d, *J* = 8.0 Hz, 1H), 7.71–7.67 (m, 4H), 4.98 (d, *J* = 8.0 Hz, 2H). ^13^C NMR (100 MHz, DMSO-*d*_6_) δ 166.95, 156.92, 156.63, 152.75, 132.37, 128.22, 127.63, 125.70, 125.17, 125.14, 125.10, 125.06, 123.94, 123.35, 123.15, 120.31, 43.63. Purity: 100%. HRMS (ESI) calculated for C_20_H_14_N_4_O_2_F_3_ [M + H]^+^: 399.1000, found: 399.1067.

5-((4-cyanobenzyl)amino)pyrimido[4,5-c]quinoline-8-carboxylic acid (**4k**)

Yellow paste (27 mg, 60%), mp: 270–275 °C; ^1^H NMR (400 MHz, DMSO-*d*_6_) δ (ppm) 13.16 (br s, 1H), 10.32 (s, 1H), 9.57 (s, 1H), 8.91 (t, *J* = 6.3 Hz, 1H), 8.72 (d, *J* = 8.3 Hz, 1H), 8.09 (d, *J* = 1.7 Hz, 1H), 7.85 (dd, *J* = 8.3, 1.7 Hz, 1H), 7.77 (d, *J* = 8.3 Hz, 2H), 7.65 (d, *J* = 8.1 Hz, 2H), 4.89 (d, *J* = 5.8 Hz, 2H). ^13^C NMR (126 MHz, DMSO-*d*_6_) δ 167.14, 156.77, 156.42, 152.98, 146.01, 145.99, 144.37, 139.04, 132.24, 132.17, 128.42, 127.57, 123.86, 123.15, 122.65, 120.43, 118.97, 109.37, 43.48. Purity: 97%, HRMS calculated for C_20_H_14_N_5_O_2_ [M + H]^+^: 356.1100, found: 356.1144.

5-(((4′-chloro-[1,1′-biphenyl]-4-yl)methyl)amino)pyrimido[4,5-c]quinoline-8-carboxylic acid (**4l**)

Pale yellow solid (45 mg, 76%), mp: 246–248 °C; ^1^H NMR (500 MHz, DMSO-*d*_6_) δ 13.16 (br s, 1H), 10.32 (s, 1H), 9.57 (s, 1H), 8.81 (t, *J* = 6.4 Hz, 1H), 8.72 (d, *J* = 8.4 Hz, 1H), 8.14 (d, *J* = 1.7 Hz, 1H), 7.85 (dd, *J* = 8.3, 1.7 Hz, 1H), 7.71–7.54 (m, 6H), 7.53–7.43 (m, 2H), 4.87 (d, *J* = 5.6 Hz, 2H).^13^C NMR (126 MHz, DMSO*-d*_6_) δ 167.18, 156.76, 156.40, 152.98, 144.57, 139.62, 139.61, 139.10, 139.07, 138.84, 137.23, 132.29, 132.09, 128.81, 128.34, 128.29, 127.56, 126.51, 123.86, 123.14, 122.49, 120.32, 43.28. Purity: 98%. HRMS (ESI) calculated for C_28_H_15_N_3_O_2_Cl [M + H]^+^: 441.1040, found: 441.1111.

5-((naphthalen-2-ylmethyl)amino)pyrimido[4,5-c]quinoline-8-carboxylic acid (**4m**)

Pale yellow solid (14.2 mg, 42%), mp: 250–253 °C; ^1^H NMR (500 MHz, DMSO*-d*_6_) δ 13.18 (br s, 1H), 10.31 (s, 1H), 9.57 (s, 1H), 8.82 (t, *J* = 6.3 Hz, 1H), 8.70 (d, *J* = 8.3 Hz, 1H), 8.13 (d, *J* = 1.7 Hz, 1H), 7.93 (s, 1H), 7.85 (ddd, *J* = 8.5, 7.1, 3.5 Hz, 4H), 7.66 (dd, *J* = 8.4, 1.7 Hz, 1H), 7.45 (qd, *J* = 7.2, 3.5 Hz, 2H), 5.00 (d, *J* = 5.6 Hz, 2H). ^13^C NMR (126 MHz, DMSO-*d*_6_) δ 167.20, 156.74, 156.36, 153.03, 144.57, 139.11, 137.53, 137.52, 132.92, 132.46, 132.09, 127.76, 127.52, 127.48, 126.33, 126.03, 125.62, 125.50, 123.87, 123.07, 122.51, 120.27, 43.80. Purity: 99%. HRMS (ESI) calculated for C_23_H_17_N_4_O_2_ [M + H]^+^: 381.1273, found: 381.1343.

5-((pyridin-2-ylmethyl)amino)pyrimido[4,5-c]quinoline-8-carboxylic acid (**4n**)

Yellow solid (27 mg, 85%), mp: 250–254 °C; ^1^H NMR (400 MHz, DMSO-*d*_6_) δ (ppm) 10.38 (s, 1H), 9.63 (s, 1H), 9.16 (br s, 1H), 8.79 (d, *J* = 12.0 Hz, 1H), 8.76–8.74 (m, 1H), 8.22 (t, *J* = 8.0 Hz, 1H), 8.17 (d, *J* = 4.0 Hz, 1H), 7.91 (dd, *J* = 4.0, 8.0 Hz, 1H), 7.86–7.84 (m, 1H), 7.69 (t, *J* = 6.5 Hz, 1H), 5.13 (d, *J* = 4.0 Hz, 2H). ^13^C NMR (212.5 MHz, DMSO-*d*_6_) δ 166.93, 156.97, 156.58, 152.81, 139.28, 132.36, 126.79, 124.25, 123.89, 123.36, 120.57, 43.49. Purity: 92%. HRMS (ESI) calculated for C_18_H_14_N_5_O_2_ [M + H]^+^: 332.1100, found: 332.1146.

5-((pyridin-3-ylmethyl)amino)pyrimido[4,5-c]quinoline-8-carboxylic acid (**4o**)

Yellow solid (26.5 mg, 81%), mp: 270–274 °C; ^1^H NMR (400 MHz, DMSO-*d*_6_) δ (ppm) 10.36 (s, 1H), 9.61 (s, 1H), 9.42 (br s, 1H), 9.07 (br s, 1H), 8.80–8.77 (m, 2H), 8.70 (d, *J* = 8.2 Hz, 1H), 8.37 (br s, 1H), 8.03–7.99 (m, 1H), 7.92 (dd, *J* = 1.8, 8.4 Hz, 1H), 5.11 (br s, 2H). ^13^C NMR (212.5 MHz, DMSO-*d*_6_) δ 166.91, 156.93, 156.56, 152.69, 144.88, 140.48, 132.45, 126.67, 123.94, 123.36, 120.48, 41.47. Purity: 92%. HRMS (ESI) calculated for C_18_H_14_N_5_O_2_ [M + H]^+^: 332.1100, found: 332.1144.

5-((pyridin-4-ylmethyl)amino)pyrimido[4,5-c]quinoline-8-carboxylic acid (**4p**)

Yellow solid (15 mg, 46%), mp: 274–276 °C; ^1^H NMR (400 MHz, DMSO-*d*_6_) δ (ppm) 10.38 (s, 1H), 9.63 (s, 1H), 9.24 (br s, 1H), 8.83–8.81 (m, 2H), 8.78 (d, *J* = 8.4 Hz, 1H), 8.13 (br s, 1H), 8.08 (d, *J* = 6.2 Hz, 2H), 7.90 (dd, *J* = 1.8, 8.4 Hz, 1H), 5.12 (d, *J* = 5.9 Hz, 2H). ^13^C NMR (212.5 MHz, DMSO-*d*_6_) δ 166.95, 156.91, 156.55, 152.91, 141.42, 132.30, 126.17, 123.94, 123.30, 120.62, 43.77. Purity: 98%. HRMS (ESI) calculated for C_18_H_14_N_5_O_2_ [M + H]^+^: 332.1100, found: 332.1145.

5-(isobutylamino)pyrimido[4,5-c]quinoline-8-carboxylic acid (**4q**)

Yellow solid (22.7 mg, 84%), mp: 270–273 °C; ^1^H NMR (500 MHz, DMSO-*d*_6_) δ (ppm) 13.10 (br s, 1H), 10.28 (s, 1H), 9.52 (s, 1H), 9.68 (d, *J* = 8.5 Hz, 1H), 8.12 (d, *J* = 1.7 Hz, 1H), 8.06 (t, *J* = 6.0 Hz, 1H), 7.82 (dd, *J* = 1.8, 8.4 Hz, 1H), 3.45 (t, *J* = 5.5 Hz, 2H), 2.14 (h, *J* = 6.8 Hz, 1H), 0.96 (d, *J* = 6.7 Hz, 6H). ^13^C NMR (125 MHz, DMSO-*d*_6_) δ 167.22, 156.64, 156.30, 153.24, 144.75, 139.02, 132.18, 127.45, 123.71, 123.94, 120.02, 47.63, 27.44, 20.80. Purity: 99%. HRMS (ESI) calculated for C_16_H_17_N_4_O_2_ [M + H]^+^: 297.1300, found: 297.1348.

5-((cyclobutylmethyl)amino)pyrimido[4,5-c]quinoline-8-carboxylic acid (**4r**)

Yellow solid (25 mg, 77%), mp: 265–268 °C; ^1^H NMR (400 MHz, DMSO-*d*_6_) δ (ppm) δ 13.10 (br s, 1H), 10.28 (s, 1H), 9.51 (s, 1H), 8.68 (d, *J* = 8.4 Hz, 1H), 8.13 (d, *J* = 1.7 Hz, 1H), 8.03 (d, *J* = 6.0 Hz, 1H), 7.82 (dd, *J* = 8.4, 1.8 Hz, 1H), 3.80–3.50 (m, 2H), 2.78 (p, *J* = 7.5 Hz, 1H), 2.03 (td, *J* = 9.0, 5.0 Hz, 2H), 1.94–1.67 (m, 4H). ^13^C NMR (100 MHz, DMSO-*d*_6_) δ 166.81, 156.21, 155.89, 152.80, 144.32, 138.57, 131.74, 127.03, 123.29, 122.63, 121.80, 119.65, 45.10, 33.87, 25.04, 17.50. Purity: 99%. HRMS (ESI) calculated for C_17_H_17_N_4_O_2_ [M + H]^+^: 309.1300, found: 309.1348.

5-((cyclopentylmethyl)amino)pyrimido[4,5-c]quinoline-8-carboxylic acid (**4s**)

Yellow solid (31 mg, 92%), mp: 261–264 °C; ^1^H NMR (400 MHz, DMSO-*d*_6_) δ (ppm) 13.09 (br s, 1H), 10.27 (s, 1H), 9.51 (s, 1H), 8.67 (d, *J* = 8.4 Hz, 1H), 8.11 (d, *J* = 1.7 Hz, 1H), 8.05 (t, *J* = 5.8 Hz, 1H), 7.82 (dd, *J* = 1.8, 8.3 Hz, 1H), 3.54 (t, *J* = 6.6 Hz, 2H), 2.42 (h, *J* = 7.3 Hz, 1H), 1.76–1.67 (m, 2H), 1.66–1.59 (m, 2H), 1.56–1.46 (m, 2H). ^13^C NMR (100 MHz, DMSO-*d*_6_) δ 167.24, 156.63, 156.28, 153.15, 144.75, 139.00, 132.15, 127.46, 123.70, 123.04, 122.19, 120.05, 45.07, 29.94, 24.77. Purity: 99%. HRMS (ESI) calculated for C_18_H_19_N_4_O_2_ [M + H]^+^: 323.1400, found: 323.1504.

5-((cyclohexylmethyl)amino)pyrimido[4,5-c]quinoline-8-carboxylic acid (**4t**)

Yellow solid (25 mg, 78%), mp: 281–283 °C; ^1^H NMR (400 MHz, DMSO *d*_6_) δ (ppm) 13.06 (br s, 1H), 10.28 (s, 1H), 9.52 (s, 1H), 8.69 (d, *J* = 8.4 Hz, 1H), 8.12 (d, *J* = 1.7 Hz, 1H), 8.05 (t, *J* = 6.0 Hz, 1H), 7.82 (dd, *J* = 1.8, 8.4 Hz, 1H), 3.48 (t, *J* = 6.4 Hz, 2H), 1.85–1.76 (m, 3H), 1.71–1.67 (m, 2H), 1.62 (br s, 1H), 1.25–1.14 (m, 3H), 1.06–0.98 (m, 2H). ^13^C NMR (100 MHz, DMSO-*d*_6_) δ 167.25, 156.65, 156.31, 153.26, 144.79, 139.04, 132.14, 127.48, 123.72, 123.07, 122.18, 120.05, 46.33, 36.97, 30.64, 26.13, 25.44. Purity: 99%. HRMS (ESI) calculated for C_19_H_21_N_4_O_2_ [M + H]^+^: 337.1600, found: 337.1660.

5-((cycloheptylmethyl)amino)pyrimido[4,5-c]quinoline-8-carboxylic acid (**4u**)

Yellow solid (32 mg, 76%), mp: 280–283 °C; ^1^H NMR (400 MHz, DMSO-*d*_6_) δ (ppm) 13.02 (br s, 1H), 10.29 (s, 1H), 9.52 (s, 1H), 8.69 (d, *J* = 8.4 Hz, 1H), 8.12 (d, *J* = 1.7 Hz, 1H), 8.08 (t, *J* = 6.0 Hz, 1H), 7.82 (dd, *J* = 1.7, 8.4 Hz, 1H), 3.47 (t, *J* = 6.5 Hz, 2H), 2.05–1.98 (m, 1H), 1.82–1.75 (m, 2H), 1.69–1.61 (br s, 2H), 1.58–1.35 (m, 6H), 1.31–1.21 (m, 2H). ^13^C NMR (100 MHz, DMSO-*d*_6_) δ 167.24, 156.66, 156.32, 153.25, 144.78, 139.03, 132.14, 127.49, 123.73, 123.07, 122.19, 120.06, 46.55, 31.75, 28.11, 25.82. Purity: 96%. HRMS (ESI) calculated for C_20_H_23_N_4_O_2_ [M + H]^+^: 351.1700, found: 351.1817.

5-(neopentylamino)pyrimido[4,5-c]quinoline-8-carboxylic acid (**4v**)

Yellow solid (12 mg, 48%), mp: 282–285 °C; ^1^H NMR (500 MHz, DMSO-*d*_6_) δ (ppm) 10.39 (s, 1H), 9.60 (s, 1H), 8.94 (br s, 1H), 8.84 (d, *J* = 8.4 Hz, 1H), 7.92 (d, *J* = 8.3 Hz, 1H), 3.92 (br s, 2H), 1.0 (s, 9H). ^13^C NMR (126 MHz, DMSO-*d*_6_) δ 166.62, 157.24, 156.96, 132.99, 124.18, 123.82, 119.86, 51.84, 33.50, 27.22. Purity: 99%. HRMS (ESI) calculated for C_17_H_19_N_4_O_2_ [M + H]^+^: 311.1400, found: 311.1503.

5-((bicyclo[2.2.1]heptan-1-ylmethyl)amino)pyrimido[4,5-c]quinoline-8-carboxylic acid (**4w**)

Yellow solid (28 mg, 77%), mp: 291–292 °C; ^1^H NMR (400 MHz, DMSO-*d*_6_) δ (ppm) 13.09 (br s, 1H), 10.29 (s, 1H), 9.53 (s, 1H), 8.69 (d, *J* = 8.4 Hz, 1H), 8.12 (d, *J* = 1.7 Hz, 1H), 8.03–7.50 (m, 2H), 3.84 (d, *J* = 5.9 Hz, 2H), 2.15 (s, 1H), 1.69–1.52 (m, 4H), 1.28 (d, *J* = 10.1 Hz, 7H). ^13^C NMR (126 MHz, DMSO-*d*_6_) δ 167.22, 156.70, 156.37, 153.26, 138.92, 132.16, 127.46, 123.71, 123.08, 122.28, 120.10, 48.78, 44.19, 41.64, 36.46, 32.66, 30.09. Purity: 100%. HRMS (ESI) calculated for C_20_H_21_N_4_O_2_ [M + H]^+^: 349.1600, found: 349.1665.

5-((((3r,5r,7r)-adamantan-1-yl)methyl)amino)pyrimido[4,5-c]quinoline-8-carboxylic acid (**4x**)

Yellow solid (19 mg, 48%), mp: 292–296 °C; ^1^H NMR (400 MHz, DMSO-*d*_6_) δ (ppm) 13.11 (br s, 1H), 10.30 (s, 1H), 9.55 (s, 1H), 8.70 (d, *J* = 8.4 Hz, 1H), 8.12 (d, *J* = 1.7 Hz, 1H), 7.98–7.36 (m, 2H), 3.42 (d, *J* = 6.2 Hz, 2H), 1.97 -1.81 (m, 3H), 1.72–1.49 (m, 12H). ^13^C NMR (100 MHz, DMSO-*d*_6_) δ 167.21, 156.72, 156.42, 153.59, 144.65, 138.88, 132.15, 127.46, 123.73, 123.10, 122.25, 120.13, 51.31, 40.20, 36.57, 34.54, 27.77. Purity: 99%. HRMS (ESI) calculated for C_23_H_25_N_4_O_2_ [M + H]^+^: 389.1900, found: 389.1979.

5-(((tetrahydrofuran-3-yl)methyl)amino)pyrimido[4,5-c]quinoline-8-carboxylic acid (**4y**)

Yellow solid (32 mg, 98%), mp: 160–164 °C; ^1^H NMR (500 MHz, DMSO-*d*_6_) δ (ppm) 13.18 (br s, 1H), 10.30 (s, 1H), 9.53 (s, 1H), 8.70 (d, *J* = 8.4 Hz, 1H), 8.30 (t, *J* = 5.9 Hz, 1H), 8.14 (d, *J* = 1.7 Hz, 1H), 7.84 (dd, *J* = 8.4, 1.7 Hz, 1H), 3.80 (td, *J* = 8.0, 5.7 Hz, 1H), 3.73 (dd, *J* = 8.5, 7.0 Hz, 1H), 3.67–3.57 (m, 5H), 2.79 (ddt, *J* = 12.9, 7.9, 3.8 Hz, 1H), 1.98 (dtd, *J* = 12.3, 8.0, 5.7 Hz, 1H), 1.72 (ddt, *J* = 12.5, 7.9, 6.4 Hz, 1H). ^13^C NMR (126 MHz, DMSO*-d*_6_) δ 167.24, 156.65, 156.33, 153.26, 144.66, 139.03, 132.42, 127.50, 123.77, 123.06, 122.36, 120.10, 70.76, 66.89, 43.17, 38.30, 29.65. Purity: 95%. HRMS (ESI) calculated for C_17_H_17_N_4_O_3_ [M + H]^+^: 325.1200, found: 325.1298.

5-(((7-oxabicyclo[2.2.1]heptan-1-yl)methyl)amino)pyrimido[4,5-c]quinoline-8-carboxylic acid (**4z**)

Yellow solid (17 mg, 43%), mp: 280–284 °C; ^1^H NMR (400 MHz, DMSO-*d*_6_) δ (ppm) 13.14 (br s, 1H), 10.31 (s, 1H), 9.54 (s, 1H), 8.72 (d, *J* = 8.4 Hz, 1H), 8.16 (d, *J* = 1.7 Hz, 1H), 7.86 (dd, *J* = 8.3, 1.8 Hz, 1H), 7.66 (t, *J* = 5.9 Hz, 1H), 4.52 (t, *J* = 5.0 Hz, 1H), 4.17–3.99 (m, 2H), 1.71 (dt, *J* = 9.3, 4.7 Hz, 1H), 1.66–1.51 (m, 6H). ^13^C NMR (100 MHz, DMSO-*d*_6_) δ 167.15, 156.80, 156.47, 153.05, 144.41, 138.74, 132.21, 127.51, 123.72, 123.17, 122.60, 120.32, 85.19, 75.85, 42.99, 32.28, 31.10. Purity: 99%. HRMS (ESI) calculated for C_19_H_19_N_4_O_3_ [M + H]^+^: 351.1400, found: 351.1454.

5-(((4,4-difluorocyclohexyl)methyl)amino)pyrimido[4,5-c]quinoline-8-carboxylic acid (**4aa**)

Yellow solid (27.8 mg, 65%), mp: 288–291 °C; ^1^H NMR (500 MHz, DMSO-*d*_6_) δ (ppm) 13.13 (br s, 1H), 10.29 (s, 1H), 9.52 (s, 1H), 8.69 (d, *J* = 8.4 Hz, 1H), 8.21 (t, *J* = 6.2 Hz, 1H), 8.13 (d, *J* = 1.7 Hz, 1H), 7.83 (dd, *J* = 8.2, 1.8 Hz, 1H), 3.55 (t, *J* = 6.5 Hz, 2H), 2.14–1.95 (m, 3H), 1.95–1.62 (m, 4H), 1.47–1.25 (m, 2H). ^13^C NMR (126 MHz, DMSO-*d*_6_) δ 167.25, 156.64, 156.33, 153.26, 144.69, 139.02, 132.33, 127.52, 126.42, 124.51, 123.76, 123.04, 122.59, 122.30, 120.06, 44.77, 44.75, 34.62, 32.63, 32.44, 32.26, 26.59, 26.51. Purity: 99%. HRMS (ESI) calculated for C_19_H_19_N_4_O_2_F_2_[M + H]^+^: 373.1400, found: 373.1474.

5-(((1-methylpiperidin-4-yl)methyl)amino)pyrimido[4,5-c]quinoline-8-carboxylic acid (**4ab**)

Yellow solid (12 mg, 52%), mp: 205–208 °C; ^1^H NMR (400 MHz, DMSO-*d*_6_) δ (ppm) 13.17 (br s, 1H), 10.32 (s, 1H), 10.24 (s, 1H), 9.55 (s, 1H), 8.73 (d, *J* = 8.4 Hz, 1H), 8.20 (s, 1H), 7.86 (dd, *J* = 8.4, 1.7 Hz, 1H), 3.56 (s, 2H), 3.34 (s, 3H), 2.99–2.84 (m, 2H), 2.67 (d, *J* = 4.6 Hz, 2H), 2.10 (d, *J* = 3.6 Hz, 1H), 1.93 (d, *J* = 14.0 Hz, 2H), 1.68–1.53 (m, 2H). ^13^C NMR (126 MHz, DMSO-*d*_6_) δ 167.05, 163.02, 156.87, 156.53, 153.04, 132.41, 123.86, 123.34, 120.07, 54.92, 53.21, 45.37, 42.55, 32.50, 27.07, 22.59. Purity: 92%. HRMS (ESI) calculated for C_19_H_21_N_5_O_2_ [M + H]^+^: 352.1700, found: 352.1770.

5-(benzyl(methyl)amino)pyrimido[4,5-c]quinoline-8-carboxylic acid (**4ac**)

Pale yellow solid (20mg, 56%), mp: 200–204 °C; ^1^H NMR (500 MHz, DMSO-*d*_6_) δ (ppm) 13.17 (br s, 1H), 10.35 (s, 1H), 9.48 (s, 1H), 8.75 (d, *J* = 8.4 Hz, 1H), 8.18 (d, *J* = 1.6 Hz, 1H), 7.89 (dd, *J* = 8.4, 1.7 Hz, 1H), 7.37 (d, *J* = 7.1 Hz, 2H), 7.32 (t, *J* = 7.5 Hz, 2H), 7.27–7.22 (m, 1H), 5.40 (s, 2H), 3.39 (s, 3H). ^13^C NMR (126 MHz, DMSO-*d*_6_) δ 167.09, 156.76, 155.37, 154.53, 143.61, 140.77, 138.76, 132.30, 128.37, 127.85, 127.56, 126.90, 125.33, 123.07, 122.70, 120.92, 55.98. Purity: 100%. HRMS (ESI) calculated for C_20_H_17_N_4_O_2_ [M + H]^+^: 345.1300, found: 345.1348.

5-((cyclohexylmethyl)(methyl)amino)pyrimido[4,5-c]quinoline-8-carboxylic acid (**4ad**)

Yellow solid (25 mg, 81%), mp: 229–231 °C; ^1^H NMR (500 MHz, DMSO-*d*_6_) δ (ppm) 13.13 (br s, 1H), 10.32 (d, *J* = 1.8 Hz, 1H), 9.50 (d, *J* = 1.5 Hz, 1H), 8.72 (dd, *J* = 8.5, 2.0 Hz, 1H), 8.15 (d, *J* = 1.8 Hz, 1H), 7.85 (dd, *J* = 8.4, 1.8 Hz, 1H), 4.09 (d, *J* = 7.1 Hz, 2H), 3.46 (s, 3H), 1.86 (ddd, *J* = 11.1, 7.6, 3.5 Hz, 1H), 1.70–1.52 (m, 5H), 1.25–1.05 (m, 3H), 0.98–0.88 (m, 2H). ^13^C NMR (126 MHz, DMSO-*d*_6_) δ 167.14, 156.66, 155.17, 154.54, 143.92, 140.83, 132.18, 127.72, 125.36, 122.66, 122.60, 120.57, 58.73, 54.89, 41.01, 36.82, 30.37, 26.02, 25.45. Purity: 98%. HRMS (ESI) calculated for C_20_H_23_N_4_O_2_ [M + H]^+^: 351.1700, found: 351.1822.

5-((1-phenylethyl)amino)pyrimido[4,5-c]quinoline-8-carboxylic acid (**4ae**)

Yellow solid (20 mg, 50%), mp: 228–232 °C; ^1^H NMR (500 MHz, DMSO-*d*_6_) δ (ppm) 13.03 (br s, 1H), 10.28 (s, 1H), 9.55 (s, 1H), 8.68 (d, *J* = 8.4 Hz, 1H), 8.27 (d, *J* = 8.2 Hz, 1H), 8.10 (d, *J* = 1.7 Hz, 1H), 7.83 (dd, *J* = 8.3, 1.7 Hz, 1H), 7.63–7.50 (m, 2H), 7.31 (t, *J* = 7.7 Hz, 2H), 7.25–7.16 (m, 1H), 5.90–4.98 (m, 1H), 1.66 (d, *J* = 7.0 Hz, 3H). ^13^C NMR (126 MHz, DMSO-*d*_6_) δ 167.16, 156.68, 156.38, 152.11, 144.88, 144.43, 138.86, 132.17, 128.24, 127.61, 126.67, 126.44, 123.77, 123.08, 122.54, 120.21, 49.43, 22.22. Purity: 97%. HRMS (ESI) calculated for C_20_H_17_N_4_O_2_ [M + H]^+^: 345.1300, found: 345.13410.

5-((2,3-dihydro-1H-inden-1-yl)amino)pyrimido[4,5-c]quinoline-8-carboxylic acid (**4af**)

Yellow solid (14.7 mg, 35%), mp: 270–273 °C; ^1^H NMR (500 MHz, DMSO-*d*_6_) δ (ppm) 13.14 (br s, 1H), 10.33 (s, 1H), 9.52 (s, 1H), 8.75 (d, *J* = 8.1 Hz, 1H), 8.18 (d, *J* = 1.8 Hz, 1H), 8.08 (d, *J* = 8.3 Hz, 1H), 7.88 (dd, *J* = 8.4, 1.9 Hz, 1H), 7.32 (dd, *J* = 10.6, 7.6 Hz, 2H), 7.23 (t, *J* = 7.4 Hz, 1H), 7.14 (t, *J* = 7.4 Hz, 1H), 6.00 (q, *J* = 7.9 Hz, 1H), 3.08 (ddd, *J* = 16.0, 8.8, 3.3 Hz, 1H), 2.93 (dt, *J* = 16.0, 8.3 Hz, 1H), 2.60 (dtd, *J* = 11.6, 7.9, 3.5 Hz, 1H), 2.24 (dq, *J* = 12.6, 8.5 Hz, 1H). ^13^C NMR (126 MHz, DMSO-*d*_6_) δ 167.20, 156.70, 156.42, 153.00, 144.59, 144.34, 143.25, 138.93, 132.36, 127.58, 127.48, 126.37, 124.62, 124.01, 123.87, 123.16, 122.60, 120.43, 55.12, 32.49, 29.94. Purity: 99%. HRMS (ESI) calculated for C_21_H_17_N_4_O_2_ [M + H]^+^: 357.1300, found: 357.1348.

(S)-5-((2,3-dihydro-1H-inden-1-yl)amino)pyrimido[4,5-c]quinoline-8-carboxylic acid (**4ag**)

Yellow solid (11.8 mg, 53%), mp: 252–255 °C; ^1^H NMR (500 MHz, DMSO-*d*_6_) δ 13.16 (br s, 1H), 10.34 (s, 1H), 9.52 (s, 1H), 8.75 (d, *J* = 8.4 Hz, 1H), 8.18 (d, *J* = 1.7 Hz, 1H), 8.10 (d, *J* = 8.3 Hz, 1H), 7.88 (dd, *J* = 8.4, 1.8 Hz, 1H), 7.38–7.29 (m, 2H), 7.23 (t, *J* = 7.3 Hz, 1H), 7.14 (t, *J* = 7.3 Hz, 1H), 6.00 (q, *J* = 7.7 Hz, 1H), 3.08 (ddd, *J* = 15.9, 8.8, 3.4 Hz, 1H), 3.01–2.87 (m, 1H), 2.59 (dtd, *J* = 12.4, 7.9, 3.4 Hz, 1H), 2.25 (ddt, *J* = 15.2, 11.0, 5.2 Hz, 1H). ^13^C NMR (126 MHz, DMSO*-d*_6_) δ 167.17, 156.70, 156.42, 153.00, 144.59, 144.36, 143.24, 138.93, 138.90, 132.32, 127.57, 127.47, 126.36, 124.61, 124.00, 123.87, 123.17, 122.59, 120.44, 55.11, 32.49, 29.95. Purity: 95%. HRMS (ESI) calculated for C_21_H_17_N_4_O_2_ [M + H]^+^: 357.1300, found: 357.1343.

(R)-5-((2,3-dihydro-1H-inden-1-yl)amino)pyrimido[4,5-c]quinoline-8-carboxylic acid (**4ah**)

Yellow solid (23.4 mg, 64%), mp: 260–262 °C; ^1^H NMR (500 MHz, DMSO-*d*_6_) δ 13.15 (br s, 1H), 10.34 (s, 1H), 9.52 (s, 1H), 8.75 (d, *J* = 8.4 Hz, 1H), 8.18 (d, *J* = 1.6 Hz, 1H), 8.10 (d, *J* = 8.3 Hz, 1H), 7.88 (dd, *J* = 8.4, 1.7 Hz, 1H), 7.32 (dd, *J* = 10.2, 7.4 Hz, 3H), 7.23 (tt, *J* = 7.4, 0.9 Hz, 1H), 7.14 (td, *J* = 7.4, 1.1 Hz, 1H), 6.00 (q, *J* = 7.8 Hz, 1H), 3.08 (ddd, *J* = 15.9, 8.8, 3.4 Hz, 1H), 2.93 (dt, *J* = 16.0, 8.2 Hz, 1H), 2.59 (dtd, *J* = 12.5, 7.9, 3.4 Hz, 1H), 2.25 (dq, *J* = 12.4, 8.4 Hz, 1H). ^13^C NMR (126 MHz, DMSO-*d*_6_) δ 167.16, 156.71, 156.42, 153.01, 144.58, 144.35, 143.24, 138.93, 132.22, 127.58, 127.47, 126.36, 124.61, 124.00, 123.86, 123.18, 122.57, 120.47, 55.11, 32.49, 32.46, 29.95. Purity: 100%. HRMS (ESI) calculated for C_21_H_17_N_4_O_2_ [M + H]^+^: 357.1300, found: 357.1343.

5-(3,4-dihydroisoquinolin-2(1H)-yl)pyrimido[4,5-c]quinoline-8-carboxylic acid (**4ai**)

Yellow solid (28.3 mg, 80%), mp: 204–208 °C; ^1^H NMR (500 MHz, DMSO*-d*_6_) δ (ppm) 13.17 (br s, 1H), 10.34 (s, 1H), 9.55 (s, 1H), 8.74 (d, *J* = 8.4 Hz, 1H), 8.24 (d, *J* = 1.7 Hz, 1H), 7.90 (dd, *J* = 8.3, 1.7 Hz, 1H), 7.28–7.24 (m, 1H), 7.19 (tt, *J* = 5.5, 2.5 Hz, 3H), 5.18 (s, 2H), 4.35 (t, *J* = 5.8 Hz, 2H), 3.11 (t, *J* = 5.8 Hz, 2H). ^13^C NMR (126 MHz, DMSO*-d*_6_) δ 167.04, 156.89, 155.65, 154.94, 143.37, 140.88, 134.84, 134.49, 132.24, 131.51, 128.58, 128.09, 126.52, 126.29, 125.94, 125.27, 123.55, 122.78, 121.24, 50.13, 46.90, 28.46. Purity: 95%. HRMS (ESI) calculated for C_21_H_17_N_4_O_2_ [M + H]^+^: 357.1300, found: 357.13464.

## 4. Conclusions

A variety of 5-benzylamino- and 5-alkylamino-substituted analogs of SGC-CK2-2 were synthesized to establish the structure-activity for CSNK2A inhibition, aqueous solubility, and antiviral activity. The 5-benzyl substituent of SGC-CK2-2 appears to occupy a lipophilic pocket that tolerates almost no polarity, since any analog with a nitrogen or oxygen atom was less active. The new analogs generally demonstrated good aqueous solubility but the CSNK2A potency and antiviral activity were not significantly improved. Analogs bearing electron withdrawing or donating substituents on the benzyl ring showed similar activity to the unsubstituted parent. In addition, replacement of the 5-benzyl group with non-aromatic amines maintained CSNK2A potency. These observations suggest that the proposed π-stacking interaction with His160 in the CSNK2A pocket was not important for binding of SGC-CK2-2 or its analogs. Conformational analysis of analogs with substituents designed to restrict the free rotation of the benzyl group supported a model in which the active analogs adopt a pose in the CSNK2A enzyme pocket close to their ground state conformation. This active conformation is consistent with the gauche twist in the 5-benzyl substituent that was observed in the X-ray cocrystal structure of SGC-CK2-2 bound to CSNK2A (PDB: 8BGC, Figure 1).

## Data Availability

Data are contained within the article.

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
