# Peer review of "Synthesis of 5-Benzylamino and 5-Alkylamino-Substituted Pyrimido[4,5-c]quinoline Derivatives as CSNK2A Inhibitors with Antiviral Activity"

_pharmaceuticals, 2024, doi:10.3390/ph17030306_

Round 1

Reviewer 1 Report

Comments and Suggestions for Authors

Authors presented valuable research on the synthesis and study of 5-amino-substituted pyrimido[4,5-c]quinolines. Deep analysis on inhibition of subunit A of casein kinase 2 by target compounds is provided. Results make a significant contribution to the design of antiviral drugs. The manuscript can be published after minor revision. Some comments:

1)      Chapter 2.1. “5-benzylamino analogs of SGC-CK2-2” is not perfect because the compound SGC-CK2-2 represent unsubstituted 5-benzylamino derivative itself. May be “counterparts of SGC-CK2-2 bearing …” would be better.

2)      In the discussion to Scheme 1 numbers of reagents and products re confusing. Why 1a, not simply 1? Second sentence “The synthesis of 5-benzylamino derivatives 4a-p…” Third sentence “…methyl ester 1a with 2-, 3-, and 4-substituted benzylamines 2a-m afforded..” Phrase below the scheme: “with amines 3a-p containing…. benzyl ring (products 4n-p, Table 3) and with aliphatic amines 2q-ab including some bridged cyclic rings (products 4q-ab, Table 4)”. The last sentence “…and constrained benzylamines 2ac-ai opening the opportunity to obtain target compounds 4ac-ai (Table 5)”.

3)      Chapter 4.5.2. Reference to the synthetic method for synthon 1a? It would be better to divide the paragraph into two: “General synthetic approach to compounds 3” and “General procedure to target products 4”. First line “… carboxylate 1a …”. Line 15 “pale-yellow solid 3. Methyl esters 3 were used in the next step without purification”. “The methyl ester product 3 …” Line 24 “The carboxylic acid product 4 …”

4)      Please check the presenting of NMR data. Do signals of COOH and NH groups look like br.s? All spectra include peak at about 13 ppm?

Reviewer 2 Report

Comments and Suggestions for Authors

This article by Willson and co-workers describes the synthesis of a collection of benzylamino- and alkylamino analogues of pyrimidio-quinoline SGC-CK2-2, a potent CSNK2A inhibitor using a previously described methodology. The synthesized compounds were evaluated regarding to their CSNK2A potency and antiviral activity against Mouse Hepatitis Virus. Additionally, aqueous solubility of all compounds and conformational constraint of a selection of the derivatives were determined.

This is a technical well-done work that provides important information about the structure-activity relation of the studies compounds in the field of CSNK2A inhibition and possible antiviral activity.

Based on these considerations, I recommend accepting this article under a MINOR revision.

Additionally, few considerations need to be considered.

  1. Page 3, 2.1. When the authors mentioned pyrimido[4,5-c]quinoline methyl ester, the number (1a) assigned in Scheme 1 for this compound is not indicated. The author have to include it in the text.
  2. Page 3, Scheme 1. This scheme is somewhat confusing. The authors attempt to summarize the synthetic methodology used, including the various types of compounds synthesized. However, it can be complicated for the reader to understand what is being indicated, especially when reading the text. Thus, the substituted benzylamines are indicated in the scheme (compounds 2a-m, 3a-m and 4a-m). However, the authors also include the 2n-p, 3n-p, and 4n-p derivatives in the scheme, which have a structure different from the generic one (they include pyridine rings). This is somewhat confusing. The authors should improve this scheme so that the variety of structures indicated can be understood. 
  3. Tables 2, 3, 4 and 5. The authors should include an additional column indicating the yields for the synthesis of each derivative.
  4. References. The abbreviations of the names of some journals indicated are not correct, and the dot has not been placed after the abbreviation. The authors should correct this.
  5. Some typing mistakes have been detected. Please, revised them. 
